# A Fluorescence-Based Histidine-Imidazole Polyacrylamide Gel Electrophoresis (HI-PAGE) Method for Rapid and Practical Lipoprotein Profiling and LDL-C Quantification in Clinical Samples

**DOI:** 10.3390/biomedicines13102560

**Published:** 2025-10-21

**Authors:** Yasuhiro Takenaka, Ikuo Inoue, Masaaki Ikeda, Yoshihiko Kakinuma

**Affiliations:** 1Department of Bioregulatory Science, Graduate School of Medicine, Nippon Medical School, Tokyo 113-8602, Japan; k12417853@nms.ac.jp; 2Department of Diabetes and Endocrinology, Saitama Medical University, Saitama 350-0495, Japan; i1901018@saitama-med.ac.jp; 3Department of Physiology, Saitama Medical University, Saitama 350-0495, Japan; mikeda@saitama-med.ac.jp

**Keywords:** lipoprotein, cholesterol, gel electrophoresis, human serum, clinical analysis

## Abstract

**Background**: Polyacrylamide gel electrophoresis (PAGE) has long been used for lipoprotein analysis, enabling the separation and profiling of lipoprotein fractions such as LDL and HDL. However, conventional disc PAGE systems are limited by low throughput and inability to directly compare multiple samples under identical conditions. Alternative methods, including high-performance liquid chromatography and agarose gel electrophoresis, require specialized equipment and expertise, limiting their clinical utility. **Methods**: We present a colorimetric and fluorescence-based histidine-imidazole PAGE (HI-PAGE) system that provides rapid, cost-effective, and reproducible separation and profiling of lipoproteins in human serum. By combining electrophoretic separation with lipid-specific fluorescent staining using Nile Red, the fluorescence-based HI-PAGE (fHI-PAGE) not only visualizes distinct migration patterns of lipoprotein fractions, but also enables the quantification of LDL-cholesterol (LDL-C). Clear resolution of LDL and other lipoprotein fractions was achieved within 1 h without band distortion, allowing for direct comparison of multiple samples on a single gel. **Results**: We validated fHI-PAGE using serum from healthy individuals and patients, demonstrating that its fluorescence-based detection was more sensitive than conventional Sudan Black B staining while providing LDL-C estimates concordant with values calculated by the Friedewald formula. Moreover, fHI-PAGE proved advantageous in cases of hypertriglyceridemia, where Friedewald calculations are unreliable. **Conclusions**: These findings establish fHI-PAGE as a practical and clinically applicable platform for simultaneous lipoprotein profiling and LDL-C quantification.

## 1. Introduction

Lipoproteins such as chylomicrons, very low-density lipoproteins (VLDLs), intermediate-density lipoproteins (IDLs), low-density lipoproteins (LDLs), and high-density lipoproteins (HDLs) are present in human blood [1]. Among these, the cholesterol contained in LDL (LDL-C) is known as bad cholesterol. Cholesterol in IDL and small dense LDL is also called “super bad cholesterol” and causes atherosclerosis and myocardial infarction [2]. Various analytical methods have been developed to predict the risk of arteriosclerosis and myocardial infarction by detecting IDL and small dense LDL. Among them, polyacrylamide gel electrophoresis (PAGE) [3,4,5,6,7,8,9,10,11], agarose gel electrophoresis [8,12,13,14,15], and chromatography [16,17] have been used since the 1960s to separate and analyze lipoproteins, and have many advantages such as low cost of equipment and reagents, easy handling, and short analysis time.

In PAGE, non-denaturing polyacrylamide disc gel analysis is frequently used to separate and evaluate serum lipoproteins. For disc electrophoresis of lipoproteins, a mixture of serum sample and staining dye is loaded onto a round glass tube filled with polyacrylamide and electrophoresis is performed [4,7,9,18,19]. This method is also used in general clinical tests. However, disc electrophoresis has limitations when comparing multiple samples under the same conditions, because only one sample can be analyzed per circular glass tube.

To solve this problem, we developed a slab-type PAGE method that can process multiple samples under the same conditions and is clinically applicable in terms of reproducibility, resolution, running time, and simple handling by adding a number of technical improvements to conventional native PAGE methods.

## 2. Materials and Methods

### 2.1. Reagents

Acrylamide, N,N′-ethylenebisacrylamide, imidazole, histidine, tris(hydroxymethyl)aminomethane (Tris), ammonium peroxodisulfate (APS), N,N,N′,N′-tetramethylethylenediamine (TEMED), Sudan Black B, Nile Red, and human serum albumin were purchased from FUJIFILM Wako Chemicals (Osaka, Japan). Cholestest^TM^ N Calibrator, used as the standard test serum (CTN), was purchased from Sekisui Medical Co., Ltd. (Tokyo, Japan) and prepared according to the manufacturer’s instructions.

### 2.2. Study Subject

This study was a non-interventional, observational study using pre-existing human serum samples collected under informed consent. No therapeutic interventions or patient follow-up were involved. Twenty-five individuals aged 40–85 years were recruited from the Department of Diabetes and Endocrinology at Saitama Medical University for this study and registered on the University Hospital Medical Information Network Clinical Trial Registry (UMIN000040373 and UMIN000050974), a non-profit organization in Japan that meets the requirements of the International Committee of Medical Journal Editors. Eligible male and female volunteers did not have a personal or family history of type 1 or 2 diabetes mellitus, pancreatitis, high blood pressure, angina pectoris, or coronary heart disease. All participants had fasting glucose levels below 110 mg/dL, body mass index below 32 kg/m^2^, total cholesterol (TC) < 220 mg/dL, triglycerides (TGs) < 200 mg/dL, smoked fewer than five cigarettes daily, and exhibited normal hepatic, thyroid, and renal functions, as determined by routine laboratory analyses. Clinical LDL-C level was calculated using the Friedewald equation:LDL-C = TC − (HDL-C + TGs/5)(1)
where the TC, TG, and HDL-C values were measured at the central laboratory of Saitama Medical University Hospital. Informed consent was obtained from all participants. This study was approved by the ethics committee of Saitama Medical University (20033.02 [approved on 16 January 2023] and 2023-021 [approved on 1 April 2024]). Sera from healthy individuals (YT and YK) for the examination of electrophoresis conditions were collected with the approval of the Ethics Committee of Nippon Medical University (A-2021-057, approved on 29 May 2021).

### 2.3. Isolation of HDL, LDL, and VLDL

HDL, LDL, and VLDL were sequentially isolated using the floating method [20,21]. Human serum from YT was mixed with an equal volume of 0.9% NaCl solution and centrifuged at 435,000× *g* for 2.5 h at 16 °C using a TL-100 tabletop ultracentrifuge, Type 42.2 Ti (Beckman Coulter, Brea, CA, USA). Lipoproteins floating in the upper part of the tube were isolated as VLDL. The bottom portion of the tube (0.5 mL) was mixed with a 16.7% NaCl solution (0.5 mL) and centrifuged at 435,000× *g* for 2.5 h. The lipoprotein floating at the top was isolated as LDL, and the bottom portion of the tube (0.5 mL) was collected as HDL.

### 2.4. Acrylamide Gel Preparation of HI-PAGE

Non-gradient uniform acrylamide gels were cast onto conventional glass plates sealed with a gasket (Nihon Eido Co., Ltd., Tokyo, Japan). We prepared a 30% acrylamide-bisacrylamide mixture (19:1) using N,N′-ethylenebisacrylamide to provide a large pore size and maintain physical gel strength (Table 1) [22]. The volume of each reagent used to prepare a single gel (90 × 70 × 1 mm, wide × height × thickness) is summarized in Table 2. First, upper and lower gel solutions, excluding TEMED, were prepared. An appropriate amount of TEMED was then added to the lower gel solution, thoroughly mixed, and poured onto the glass plates. Next, the lower gel solution was gently overlaid with distilled water and left to stand at 25 °C for at least 10 min. After the lower gel solution solidified, the overlying distilled water was completely removed. The upper gel solution was mixed with TEMED, poured, and the comb was inserted. The mixture was then incubated at 25 °C for more than an hour. To prevent over-drying, the gel was wrapped in plastic, and stored overnight in a refrigerator.

### 2.5. Running Buffers for HI-PAGE

The preparation of the HI-PAGE running buffer is presented in Table 3. Weighed Tris and histidine were dissolved in an appropriate amount of ultrapure water, mixed thoroughly, then stored at 4 °C. pH adjustment was not necessary; however, when the buffer was prepared as shown in Table 2, the pH was approximately 8.4. The amount of HI-PAGE running buffer to be prepared was adjusted according to the size of the electrophoresis tank.

### 2.6. Preparation of the Pre-Staining Solution

Sudan Black B powder (0.3 g) was dissolved in 10 mL of dimethyl sulfoxide (DMSO) to prepare a 3% (*w*/*v*) Sudan Black DMSO solution, which was stored at room temperature. However, as the staining properties gradually decrease, the solution should be prepared every 2–3 months. The pre-staining solution was prepared by mixing the components listed in Table 4 from top to bottom. It is important to follow this sequence, because altering it can result in an incomplete dispersion of Sudan Black B throughout the solution, causing it to precipitate as insoluble granules. Each component was mixed vigorously upon addition. The pre-staining solution was stored in a refrigerator and prepared every 2–3 months because of the gradual decrease in its staining properties. For the fluorescence pre-staining solution, Nile Red, Tris buffer, bromophenol blue, and glycerol were mixed, as shown in Table 5.

### 2.7. Sample Preparation

Before loading the samples onto the upper gel, the human serum was pre-stained using the solution listed in Table 5. The volume ratio of the sample to the pre-staining solution was adjusted according to the concentration and total amount of lipoproteins in the sample, with a typical ratio of 1:1 (sample:pre-staining solution). Human serum from healthy individual (YT) was used in this study (Figure 1).

### 2.8. Sample Application and Electrophoresis Conditions

After mixing the sample with the pre-staining solution, the mixture was incubated at room temperature for 5 min to allow the Sudan Black B or Nile Red to bind to lipoproteins in the sample. The entire sample volume (typically 5 µL) was then loaded into each well of the upper gel. Loading was performed slowly and carefully using loading tips. Once the stained samples were loaded, electrophoresis was performed at a constant current (20–40 mA) at ambient temperature (~22–25 °C). The duration of electrophoresis depended on the size of the gel; a gel measuring 90 × 70 × 1 mm (w × h × t) was run for approximately 30–40 min at 40 mA.

### 2.9. Detection of Lipoproteins and Densitometric Quantification

After electrophoresis, gel images were obtained either by photographing the gel or by capturing the fluorescence of lipoproteins using a laser scanner, while the gel remained sandwiched between gel plates. Due to the soft nature of the gel, handling it after its removal from the gel plates was challenging. To obtain fluorescence images of HI-PAGE, ChemiDoc Touch (Bio-Rad, Hercules, CA, USA) or FLA 7000 (with excitation at 473 nm and filter O580; GE Healthcare, Chicago, IL, USA) was used. For quantification of LDL smear intensity, a fixed rectangular ROI was manually drawn over the smeared LDL region based on visual inspection, and the integrated density within the ROI was measured using Fiji (ImageJ2, Ver. 2.9.0). The same ROI size and position were applied across samples. The standard curve for LDL quantification was constructed in GraphPad Prism (ver. 5) using a nonlinear regression model. The curve was fitted by the least squares (ordinary) method, and the goodness of fit (R^2^) was approximately 0.96–0.98. This equation was also applied to calculate LDL amounts in serum samples using the “Interpolate unknowns from standard curve” function in the same Prism file.

### 2.10. Disc PAGE

Conventional disc gel electrophoresis, Lipophor AS, was performed at the ASKA Special Inspection Laboratory (Kawasaki, Japan) (https://aska-labo.com/) (accessed on 31 July 2025). The corresponding result was scanned from the clinical analysis report and is shown as Figure 2H.

## 3. Results

### 3.1. Colorimetric Analysis of Human Lipoproteins Using HI-PAGE

HI-PAGE analysis of human serum from healthy individuals revealed that the use of Tris(hydroxymethyl)aminomethane hydrochloride as a buffer in the upper gel and imidazole hydrochloride in the lower gel, combined with Tris-histidine running buffer, effectively prevented band distortions of lipoproteins within an hour of electrophoresis (Figure 1A). Thus, this method guaranteed the precise analyses of multiple samples containing lipid–protein complexes, minimizing the occurrence of smiling and/or smearing features even within a shorter running time. In contrast, traditional non-denaturing polyacrylamide slab gel analysis using Tris buffer for the lower gel and Tris-glycine running buffer resulted in smiling or smearing band profiles of HDLs (Figure 1B,C). Moreover, band separation between VLDLs and LDLs using the traditional method was insufficient (Figure 1B) compared with the HI-PAGE results (Figure 1A), which required a longer electrophoresis time.

### 3.2. Fluorescence Analysis of Human Lipoproteins Using HI-PAGE

To further enhance the sensitivity of HI-PAGE, we pre-stained the standard test serum (CTN) with fluorescent dyes using Nile Red [23,24] as alternatives to Sudan Black B before HI-PAGE electrophoresis. At the end of the run, the gel was immediately visualized using laser scanning or a fluorescent CCD imager, without separating the gel from the glass plates. As shown in Figure 2A, the electrophoresis images of the VLDLs and LDLs stained with Nile Red were comparable to those obtained by colorimetric HI-PAGE (Figure 1A). We also obtained standard curves for VLDLs (Figure 2B) and LDLs (Figure 2C) by quantifying band intensities in the fluorescence HI-PAGE (fHI-PAGE) gel images. However, upon examining the fluorescence profile of purified human albumin, the most abundant protein in the serum, using fHI-PAGE, we observed a notable signal with mobility similar to that of HDL in the gel (Figure 2D, a band indicated by an asterisk in lane 2), presumably due to bilirubin fluorescence that binds to human albumin [25]. The signal at the HDL position was not observed in the human albumin sample in the colorimetric HI-PAGE analysis (Figure 2E, lane 2), indicating that the band observed in HI-PAGE (Figure 1A–C and Figure 2E, lane 1) was derived from HDLs. We attempted to separate HDLs and albumin bound to the fluorescent substance by increasing the acrylamide concentration (6% lower acrylamide concentration) without success (Figure 2F). Therefore, to specifically detect HDLs using fHI-PAGE, further improvements are needed such as optimization in the fluorescent dye used or a reduction in the fluorescent signal from the fluorescent substance. Nonetheless, the current fHI-PAGE method is suitable for the qualitative and quantitative analysis of LDLs and VLDLs because no detectable signal was observed at the position corresponding to LDLs and VLDLs in the human albumin sample (Figure 2D, lane 2). The fHI-PAGE method was applied to the analysis of clinical samples (Figure 2G), resulting in lipoprotein profiles similar to those obtained by conventional disc electrophoresis (Figure 2H).

### 3.3. Fluorescence Analysis of LDL in Clinical Samples Using HI-PAGE

We applied fHI-PAGE to clinical samples from 18 patients to evaluate the LDL-C levels. To calibrate LDL-C, we used varying amount of CTN along with patient serum samples for fHI-PAGE (Figure 3A,B). Densitometry was performed on the LDL signal following laser scanning of the gel, and LDL-C was calculated using CTN standards. We then compared the LDL-C values obtained using fHI-PAGE with those calculated using the Friedewald equation (n = 22) and observed a moderate correlation between the two methods (R^2^ = 0.507) (Figure 3C). To understand the discrepancies between these methods, we focused on several outlier samples. For example, the LDL-C values of Patient B were 284 mg/dL and 78 mg/dL when calculated using the fHI-PAGE and Friedewald equation, respectively (Figure 3C,D). This discrepancy is likely due to Patient B’s high triglyceride level (328 mg/dL), which led to a significantly lower LDL-C value when calculated using the Friedewald equation. In contrast, the LDL-C value for Patient G calculated using the Friedewald equation was 250 mg/dL, which was notably higher than that obtained using fHI-PAGE analysis (198 mg/dL). This may be attributed to the high total cholesterol level of Patient G (340 mg/dL). On the other hand, LDL-C could not be calculated using the Friedewald equation for Patient J because of an abnormally high triglyceride level (2048 mg/dL), whereas the LDL-C level was successfully measured using fHI-PAGE analysis (Figure 3D).

## 4. Discussion

We developed a rapid, simple, sensitive, and high-throughput slab-type PAGE system for lipoprotein analysis. This technology is based on the native PAGE method for lipoproteins, including conventional disc gel electrophoresis, with several modifications such as buffer systems and dyes to stain lipoproteins. We speculate that the improved separation between VLDL and LDL observed in Figure 1A compared with Figure 1B can be primarily attributable to the use of histidine in the electrophoresis running buffer (Tris-histidine, pH 8.4) as opposed to glycine (Tris-glycine, pH 8.3) employed in the conventional method. At this pH, histidine is more negatively charged than glycine and likely migrates more efficiently toward the anode, thereby contributing to the sharper and faster migration of lipoprotein bands. Compared with disc electrophoresis, slab electrophoresis allows for a simpler operation as it eliminates the need to prepare multiple tube gels for handling multiple samples. In addition, it enables sample comparison under identical conditions on the same gel, which is advantageous for detecting subtle differences between the samples. Furthermore, because there were multiple wells, we obtained a calibration curve by preparing standard serum at several amounts, ranging from the upper to lower limits (Figure 2A–C). In addition, whereas conventional disc PAGE requires 25 µL of serum for analysis, 5 and 1 µL are sufficient for HI-PAGE and fHI-PAGE analyses, respectively. Conventional disc PAGE requires photopolymerization of the loading gel for 30–45 min before electrophoresis, whereas our method allows for the immediate application after mixing serum and staining dye.

In conventional slab-type PAGE, the bands of each lipoprotein are prone to distortion due to smearing or smiling during electrophoresis. Here, we improved the distortion of the band shapes of VLDLs, LDLs, and HDLs by using imidazole as the lower gel buffer and Tris-histidine as the running buffer. Furthermore, the slab-type PAGE reported by Singh et al. requires a total run time of 190 min [3], whereas in the HI-PAGE system, electrophoresis can be completed within 40 min, which is almost the same as that of conventional disc PAGE.

Our method had some limitations: (1) limited linearity when the signal intensities of lipoproteins were scanned using a densitometer, especially at a higher amount of LDLs; (2) a notable mobility shift of LDLs was observed when increasing the loading amount, which caused difficulties in analyzing the slight mobility change of LDLs between samples with different loading amounts; and (3) as shown in Figure 2D, HDLs were not measured accurately using fHI-PAGE. Nevertheless, this method has numerous advantages including affordability, simplicity, and the ability to perform real-time monitoring during analysis without expensive equipment such as high performance liquid chromatography [26] or other specialized analytical techniques. We believe that further improvements and applications of this method have a significant potential.

## 5. Conclusions

In conclusion, we developed a colorimetric and fluorescent slab PAGE method with high-throughput, cost-effective, simple, rapid, and reproducible analysis of lipoproteins in the human serum using Tris-histidine as the running buffer and imidazole buffer for the polyacrylamide gel (HI-PAGE). Fluorescent HI-PAGE was applied to the clinical samples, revealing that the method is highly sensitive and allows for the quantitative detection of lipoproteins in human serum.

## Figures and Tables

**Figure 1 biomedicines-13-02560-f001:**
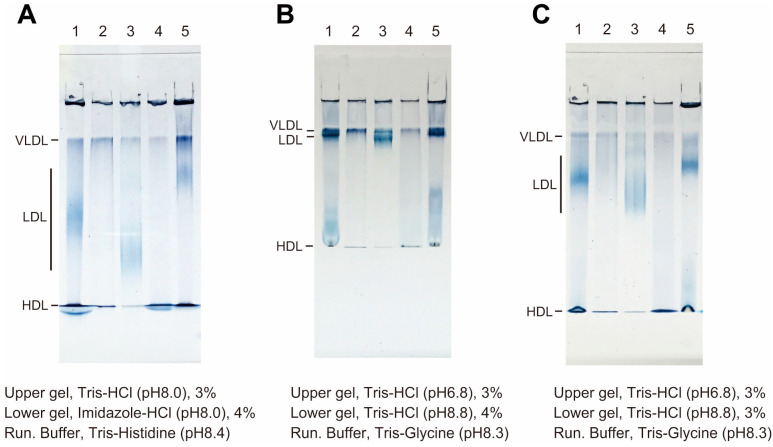
Electrophoresis of human lipoproteins using histidine-imidazole polyacrylamide gel electrophoresis (HI-PAGE). (**A**) Serum, VLDLs, LDLs, and HDLs from healthy individual (lanes 1–4, respectively) and standard test serum (lane 5) were analyzed using colorimetric HI-PAGE. Compositions of the upper, lower, and running buffers are indicated below the gels. The positions of VLDLs, LDLs, and HDLs are shown on the left side of the gel. (**B**) Electrophoretic profiles of conventional native PAGE with Tris-glycine running buffer. The percentage of lower gel was 4%. (**C**) The percentage of lower gel was 3%.

**Figure 2 biomedicines-13-02560-f002:**
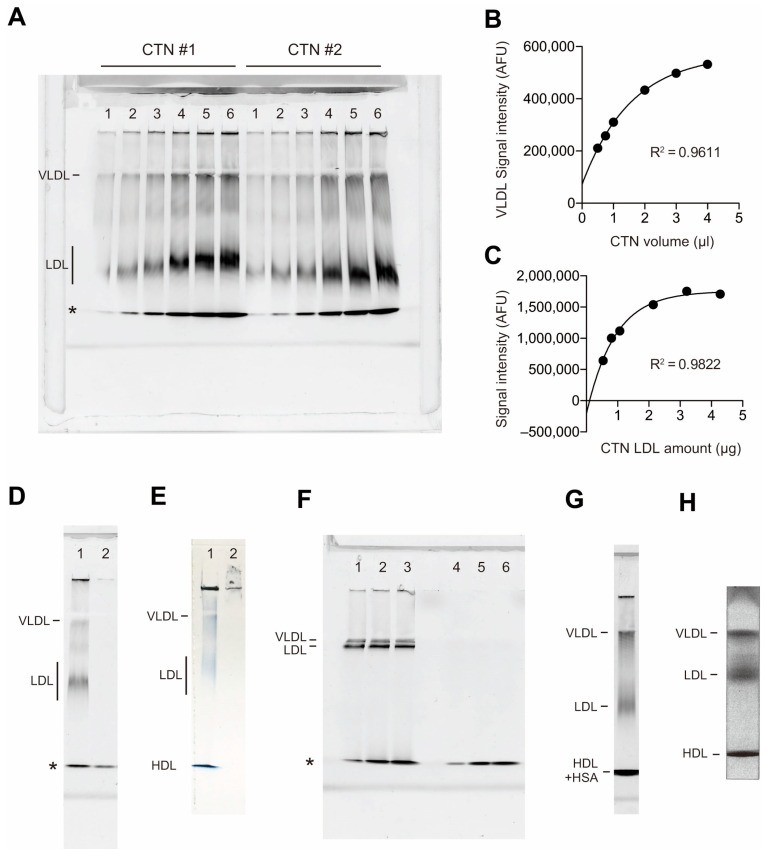
Electrophoresis of human lipoproteins using fluorescent HI-PAGE (fHI-PAGE). (**A**) Different amounts of standard test serum (CTN) were pre-stained with Nile Red and applied onto the gel (n = 2), and the fluorescent signal was imaged using laser scanning. Asterisks indicate bands that comigrate with HDL but likely represent bilirubin fluorescence from albumin, rather than HDL itself. Standard curves for VLDLs (**B**) and LDLs (**C**). The volumes of CTN loaded were 0.5, 0.75, 1, 2, 3, and 4 µL, corresponding to lanes 1–6 in (**A**). (**D**) Human serum albumin (HSA) shows fluorescence at the size of HDL (asterisk). Lane 1, CTN; lane 2, human serum albumin. (**E**) Colorimetric HI-PAGE does not show a signal corresponding to the size of HDL. Lane 1, CTN; lane 2, human serum albumin. (**F**) fHI-PAGE analysis of different amounts of CTN (lanes 1–3) and HSA (lanes 5–7) with 6% acrylamide in the lower gel. An asterisk indicates the size of HDL. (**G**) fHI-PAGE analysis of the clinical sample. (**H**) Colorimetric disc electrophoresis of the same sample used in (**G**). The sample was tested by the ASKA Special Inspection Laboratory.

**Figure 3 biomedicines-13-02560-f003:**
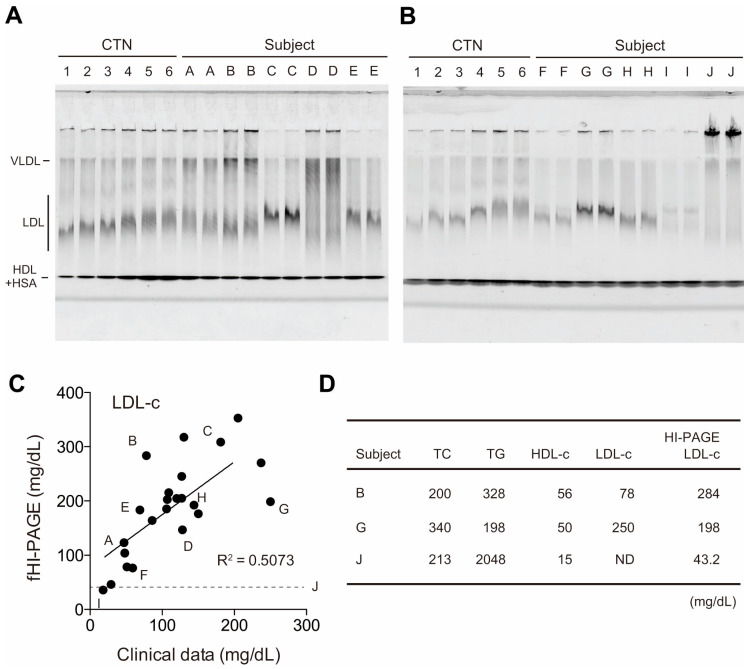
fHI-PAGE analysis of clinical samples. (**A**) CTN (lanes 1–6) and clinical samples (lanes A–E, duplicates) were analyzed using fHI-PAGE. (**B**) Additional gel image of fHI-PAGE with CTN (lanes 1–6) and clinical samples (lanes F–J, duplicates). (**C**) Dot plot of values obtained using densitometry analysis of the LDL signal of fHI-PAGE (*y* axis) and that calculated using the Friedewald equation (*x* axis). The letters correspond to those in (**A**,**B**). (**D**) Total cholesterol (TC), triglyceride (TG), HDL-C, and calculated LDL-C values with the fHI-PAGE LDL-C results of the outfitter samples. The unit of each value is mg/mL.

**Table 1 biomedicines-13-02560-t001:** Composition of 30% acrylamide-bisacrylamide mixture (19:1).

		Unit
Acrylamide	11.4	g
N,N′-ethylenebisacrylamide	0.6	g
Ultrapure water	Appropriate volume	mL
Total	40	mL

**Table 2 biomedicines-13-02560-t002:** Composition of the upper and lower gels in HI-PAGE.

	Upper Gel		Lower Gel	
	Volume (mL)	Concentration	Volume (mL)	Concentration
1 M Tris-HCl, pH 8.0	0.375	0.125 M	0	
1 M imidazole-HCl, pH 8.0	0		2.40	0.370 M
30% acrylamide-bisacrylamide mixture (19:1)	0.300	3.00%	0.870	4.00%
Ultrapure water	2.33		3.14	
10% APS	0.0380		0.0820	
TEMED	0.00450		0.0100	
Total	3.05		6.50	

**Table 3 biomedicines-13-02560-t003:** Composition of the HI-PAGE running buffer.

		Final Concentration
Tris	0.91 g	0.025 M
Histidine	6.0 g	0.13 M
Ultrapure water	up to 300 mL	

**Table 4 biomedicines-13-02560-t004:** Composition of the pre-staining solution.

	Volume (µL)	Final Concentration
3% Sudan Black DMSO solution	40	0.12%
10% DDM	10	0.10%
80% glycerol	250	20%
1 M Tris-HCl, pH 7.4	50	0.05 M
Ultrapure water	650	
Total	1000	

**Table 5 biomedicines-13-02560-t005:** Composition of the fluorescence pre-staining solution.

	Volume (µL)	Final Concentration
20 mg/mL Nile Red DMSO solution	25	0.5 mg/mL
1% bromophenol blue	37.5	0.0375%
80% glycerol	625	50%
1 M Tris-HCl, pH 8.0	50	0.05 M
Ultrapure water	262.5	
Total	1000	

## Data Availability

Data supporting the findings of this study are available from the corresponding author upon reasonable request.

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
