# Peer review of "A Fluorescence-Based Histidine-Imidazole Polyacrylamide Gel Electrophoresis (HI-PAGE) Method for Rapid and Practical Lipoprotein Profiling and LDL-C Quantification in Clinical Samples"

_biomedicines, 2025, doi:10.3390/biomedicines13102560_

Round 1
Reviewer 1 Report
Comments and Suggestions for Authors
1- The type of study must be noted.
2- Despite of positive net charge of Imidazole in lower gel 1A, why were the VLDL, LDL bands with negative net charge separated better than 1(B) ? please explain?
3- The most of references are old? more new references are needed to use.
Author Response
Comments 1:
The type of study must be noted.
Our Response 1:
We thank the reviewer for this important comment.
In response, we have clarified the type of study at the beginning of section 2.2. Study subject in the Materials and Methods (Page 7, Lines 78-80). Specifically, we have added the following sentence to clearly describe the observational nature of the study:
“This study is a non-interventional, observational study using pre-existing human serum samples collected under informed consent. No therapeutic interventions or patient follow-up were involved.”
This addition helps to clarify that the study was conducted without any clinical intervention and was based on previously collected samples, in accordance with ethical standards.
Comments 2:
Despite of positive net charge of Imidazole in lower gel 1A, why were the VLDL, LDL bands with negative net charge separated better than 1(B) ? please explain?
Our Response 2:
Thank you very much for your insightful comment.
We have added an explanation to the Discussion section to address this point (Page 16, Lines 232-237). In the revised manuscript, we state:
“We speculate that the improved separation between VLDL and LDL observed in Fig. 1A, compared to Fig. 1B, is primarily attributable to the use of histidine in the electrophoresis running buffer (Tris-histidine, pH 8.4), as opposed to glycine (Tris-glycine, pH 8.3) employed in the conventional method. At this pH, histidine is more negatively charged than glycine and likely migrates more efficiently toward the anode, thereby contributing to sharper and faster migration of lipoprotein bands.”
As described above, we consider the main contributing factor to be the increased electrophoretic mobility of histidine, which may enhance the overall current flow and lipoprotein separation efficiency. We also note that the imidazole-HCl used in the lower gel (pH 8.0) could have a minor supplementary effect, possibly by modulating the local buffering environment, although its specific contribution remains to be determined.
Comments 3:
The most of references are old? more new references are needed to use.
Our Response 3:
Thank you for pointing out the need to include more recent literature. While this electrophoretic approach is rooted in longstanding methodology, we agree that including modern references strengthens the context. Accordingly, we have added two recent studies to the revised manuscript.
Qiao et al. (2022), in which gradient gel electrophoresis is reviewed as a current method for LDL subclass separation.
Behling-Kelly et al. (2022), using slab agarose gel electrophoresis for bovine lipoprotein profiling.
We believe inclusion of these recent analyses helps to situate our proposed fHI-PAGE method within the evolving landscape of lipoprotein and lipid separation technologies, demonstrating that gel- and electrophoresis-based strategies remain of interest in current research.
Reviewer 2 Report
Comments and Suggestions for Authors
The author described an improved method in quantification of lipoproteins in human serum sample from clinical sources. The method is innovative and can be a significant improvement over the current methods. I have the following concerns:
- Please describe the quantification with more detail especially dealing with the smearing LDL quantification. Where the boundary is for the LDL smear, what analysis software is used and how the threshold is defined for the LDL smear in the software?
- In figure 2, the standard curve was derived using LDL and signal intensity. The standard curve is non-linear. Is there any effort to linearize the standard curve? Or the standard curve is considered linear when the serum loading is low enough? Please describe in more detail. What is the equation applied in real clinical samples?
- In addition, does the quantification method described fit well in simpler samples or even a pure LDL samples with know concentration?
Author Response
Comments 1:
Please describe the quantification with more detail especially dealing with the smearing LDL quantification. Where the boundary is for the LDL smear, what analysis software is used and how the threshold is defined for the LDL smear in the software?
Our Response 1:
We appreciate the reviewer's thoughtful question regarding the quantification of LDL smearing. As noted, in cases where LDL appears as a continuous smear rather than a distinct band, it was challenging to define a strict threshold or boundary by automated analysis. Therefore, we performed densitometric quantification using Fiji (ImageJ2, Ver. 2.9.0) by manually defining a consistent region of interest (ROI) for each smear. The ROI boundaries were carefully set based on visual inspection of the band pattern and were applied uniformly across all relevant lanes to ensure comparability.
We acknowledge that this method involves a degree of subjectivity, but we minimized variability by applying the same ROI size and location to each sample lane, and by performing the quantification in a blinded manner with respect to sample identity. We have now clarified this procedure in the Methods section (2.9. Detection of Lipoproteins and Densitometric Quantification) as follows:
“For quantification of LDL smear intensity, a fixed rectangular ROI was manually drawn over the smeared LDL region based on visual inspection, and the integrated density within the ROI was measured using Fiji (ImageJ2, Ver. 2.9.0). The same ROI size and position were applied across samples.”
Comments 2:
In figure 2, the standard curve was derived using LDL and signal intensity. The standard curve is non-linear. Is there any effort to linearize the standard curve? Or the standard curve is considered linear when the serum loading is low enough? Please describe in more detail. What is the equation applied in real clinical samples?
Our Response 2:
Thank you for your valuable comment. As shown in Figure 2B, the standard curve between LDL amount and signal intensity was fitted using a non-linear regression model (One phase decay) in GraphPad Prism ver.5. The equation used for the fitting was:
Y = (Yâ‚€ − Plateau) × exp(−k × X) + Plateau
where Y is the signal intensity and X is the LDL amount. The fit was performed using ordinary least squares, and the resulting R² value was approximately 0.96-0.98, indicating a good fit.
This equation was also used to interpolate the LDL amounts in unknown serum samples using the “Interpolate unknowns from standard curve” function in the same Prism file. We did not apply any additional transformation (e.g., linearization) since the non-linear model captured the relationship well across the measured range.
We have added this information to the Materials and Methods section (page 12, lines 159–163).
“The standard curve for LDL quantification was constructed in GraphPad Prism (ver.5) using a non-linear regression model. The curve was fitted by the least squares (ordinary) method, and the goodness of fit (R²) was approximately 0.96-0.98. This equation was also applied to calculate LDL amounts in serum samples using the “Interpolate unknowns from standard curve” function in the same Prism file.”
Comments 3:
In addition, does the quantification method described fit well in simpler samples or even a pure LDL samples with know concentration?
Our Response 3:
We appreciate the reviewer's question regarding whether our quantification method is applicable to simpler or purified LDL samples.
While we have applied the HI-PAGE method to analyze commercially available LDL and ultracentrifugation-purified LDL samples—as shown in Figure 1—we did not generate the standard curve using these purified LDL samples. Instead, the standard curve was constructed from serial dilutions of standard test serum with estimated LDL contents, allowing us to interpolate unknown LDL values from test samples using Prism software.
That said, the clear banding patterns observed with purified LDL support the idea that our quantification method is fundamentally applicable to simpler systems as well. However, our focus in this study was on assessing LDL quantities in complex biological samples such as serum.